# IF-Font: Ideographic Description Sequence-Following Font Generation

**Xinping Chen, Xiao Ke,**∗**Wenzhong Guo**

Fujian Provincial Key Laboratory of Networking Computing
and Intelligent Information Processing,
College of Computer and Data Science, Fuzhou University, Fuzhou 350116, China
Engineering Research Center of Big Data Intelligence,
Ministry of Education, Fuzhou 350116, China
{221027017, kex, guowenzhong}@fzu.edu.cn

## Abstract

Few-shot font generation (FFG) aims to learn the target style from a limited number of reference glyphs and generate the remaining glyphs in the target font. Previous works focus on disentangling the content and style features of glyphs, combining the content features of the source glyph with the style features of the reference glyph to generate new glyphs. However, the disentanglement is challenging due to the complexity of glyphs, often resulting in glyphs that are influenced by the style of the source glyph and prone to artifacts. We propose IF-Font, a novel paradigm which incorporates Ideographic Description Sequence (IDS) instead of the source glyph to control the semantics of generated glyphs. To achieve this, we quantize the reference glyphs into tokens, and model the token distribution of target glyphs using corresponding IDS and reference tokens. The proposed method excels in synthesizing glyphs with neat and correct strokes, and enables the creation of new glyphs based on provided IDS. Extensive experiments demonstrate that our method greatly outperforms state-of-the-art methods in both one-shot and few-shot settings, particularly when the target styles differ significantly from the training font styles. The code is available at https://github.com/Stareven233/IF-Font.

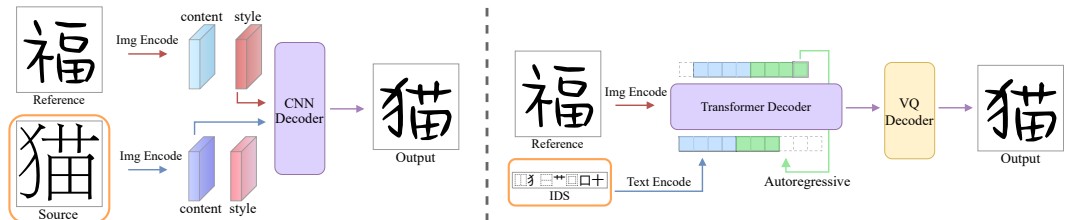

Figure 1: Comparison of two font generation paradigms. **Left:** The style-content disentangling paradigm. It assumes that a glyph can be decomposed into two distinct attributes: content and style. **Right:** The proposed paradigm. We first autoregressively predict the target tokens and decode them with a VQ decoder. Orange boxes show the main difference between the two paradigms.

---

∗Corresponding author

38th Conference on Neural Information Processing Systems (NeurIPS 2024).

# 1  Introduction

At the heart of font generation lies the extraction of styles from some reference glyphs of a certain font, and generate the remaining glyphs of this font. Some languages, such as Chinese, Japanese, and Korean, have a large number of characters and intricate glyph structures. Font generation can significantly reduce the labor intensity of font designers and support tasks like handwriting imitation, ancient book restoration, and internationalization of film and television productions.

EMD [58] and SA-VAE [44] are based on the belief that the target glyph can be generated by integrating the content features of the source glyph with the style features of the reference glyph, as illustrated in Fig. 1 (left). The majority of subsequent works [53, 38, 47, 25, 50] continues this paradigm, but this makes font generation a sub-task of image-to-image translation, where the source glyph is morphed to match the style of the reference glyphs, rather than being truly "generated". Due to the complex structure of glyphs, achieving a distinct boundary between style and content features requires considerable effort. Consequently, glyphs produced through the disentangling strategy typically maintain similar stroke thickness to the reference glyph but align more closely with the content font regarding spatial structure, size, and inclination.

To this end, DG-Font [54] incorporates deformable convolution. Diff-Font [15] integrate diffusion process to improve the network's learning capabilities. CF-Font [50] proposes content fusion. Additionally, several approaches [38, 47, 25, 30] combine fine-grained prior information such as strokes and components, to further enhance generation quality. These methods essentially follow the content-style disentanglement paradigm, in scenarios where the content font differs substantially from the target font, the resulting glyphs are susceptible to artifacts such as missing strokes, blur, and smudge.

We abandon the source glyphs in favor of Ideographic Description Sequence (IDS) to convey content information. It is based on a simple fact: without disentangling features, there is no risk of incomplete disentanglement. Consequently, font generation is reframed as sequence prediction task, where the objective is to generate the tokens of the target glyph based on the given IDS and reference glyphs. This approach mitigates the impact of source glyphs on the outputs and diminishes artifacts by leveraging the prior knowledge embedded in the quantized codebook. The users are allowed to formulate IDSs to create non-existing Chinese characters, such as kokuji[2] (Japanese-invented Chinese characters), provided that the corresponding structures and components have been learned during training. This endows the model with certain cross-linguistic capabilities. We refer to this method as **I**deographic Description Sequence-**F**ollowing **Font** Generation, or IF-Font. In summary, the key contributions of this paper are as follows:

- We propose IF-Font, which abandons the previous content-style disentanglement paradigm and generates glyphs through next-token prediction.
- We devise a novel IDS Hierarchical Analysis (IHA) module that analyzes the spatial structures and components of Chinese characters. It allows our decoder flexibly control the generated content with the encoded semantic features.
- Leveraging corresponding IDSs, we design the Structure-Style Aggregation (SSA) block to extract and efficiently aggregate the style features of reference glyphs.

# 2  Related Works

**Image-to-image translation**    Image-to-image translation (I2I) aims to learn a mapping from a source domain to a target domain, requiring the transformation of images in the source domain into those in the reference style's target domain while preserving their content. Pix2Pix [21] is the first I2I method that trains GANs [13] using paired data. CycleGAN [60] achieves unsupervised training through cycle consistency loss, although it is limited to transformations between two domains. UNIT [29] enforces the latent codes of images from two distinct domains to be identical, while employing separate generators for images in each domain. This process embodies the concept of disentanglement. MUNIT [19] further refines UNIT's latent code into content and style codes. Multimodal image translation can be achieved by combining the content code with different style codes.

---

[2] https://www.sljfaq.org/afaq/kokuji-list.html

While applying an image-to-image translation framework for font generation is currently the mainstream approach, we believe this to be inappropriate. Unlike ordinary images, the boundaries between content and style in glyphs are ambiguous. For example, although handwriting will certainly differ when the same characters are written by different writers, their semantic meanings remain unchanged. Given that glyph features are challenging to disentangle, we utilize style-neutral IDSs to determine characters, thus avoiding any influence on the styles of results due to insufficient disentanglement of content glyphs.

**Few-shot font generation**  Few-shot learning [12] represents the prevailing research focus in font generation, aiming to simulate the target style with just a handful of reference glyphs. Font generation methods fall into two categories based on their utilization of implicit structural information within glyphs. Methods treating glyphs as general images possess broader applicability, enabling generation across various languages. Conversely, methods leveraging structural information typically yield higher quality outputs but are confined to specific language.

Among the methods that do not incorporate structural information, EMD [58] stands out as the earliest attempt to disentangle glyphs into content and style features. DG-Font [54] employs deformable convolution [7] to capture the glyph deformations. FontRL [31] uses reinforcement learning [45] to draw the skeleton of Chinese characters. FontDiffuser [55] models the font generation task as a noise-to-denoise paradigm. Shamir et al. [42] explores a parametric representation of oriental alphabets, which can elegantly balance glyph quality and compression. In vector font generation, Deepvecfont [51] exhaustively exploit the dual-modality information (raster images and draw-command sequences) of fonts to synthesize vector glyphs. CLIPFont [43] controls the desired font style through text description rather than relying on style reference images.

In terms of methods that incorporate structural information, SA-VAE [44] utilizes the radicals and spatial structures of Chinese characters. CalliGAN [53] adopts the Zi2Zi framework [48] and fully decompose Chinese characters into sequences of components. SC-Font [23] further decomposes Chinese characters into stroke granularity. DM-Font [4] proposes dual memory to update component features continuously. LF-Font [38] represents component-wise style through low-rank matrix factorization [3]. MX-Font [37] automatically extracts the features through multiple localized experts. FS-Font [47] demands that reference glyphs include all components of the target glyph, otherwise may result in a degradation of generation quality. CG-GAN [25] employs GRU [6] and attention mechanism to predict component sequences. XMP-Font [30] performs multimodal pre-training on Chinese character strokes and glyphs data. Most of the above methods are constrained by the content-style disentanglement paradigm. They often neglect the presence of Ideographic Description Character (IDC) which refers to the spatial structure of Chinese characters, suffering from artifacts and inconsistent styles.

**Vector quantized generative models**  Vector Quantization (VQ) typically follows a two-stage training scheme. Initially, it employs a codebook to record and update vectors, converting them from a continuous feature space to a discrete latent space. Subsequently, it models the distribution of these quantized vectors with a decoder to predict tokens, which are the codebook indices, and then restores the tokens to a image.

VQ-VAE [35] is the first to incorporate quantization into the variational autoencoder (VAE) [24] framework. VQ-VAE2 [41] performs multi-scale quantization and adopts rejection sampling [1] VQGAN [9] acquires the codebook with the help of GAN [13] and employs Transformer [49] to replace the PixelCNN [34] used by VQ-VAE [35]. RQ-VAE [26] proposes a residual quantizer. BEiT [2] performs masked image modeling (MIM) on the patch view with the supervision of visual tokens. MaskGIT [5] directly models visual tokens and proposes parallel decoding. MAGE [27] is similar to MaskGIT [5], but introduces ViT [8] and contrastive learning [14]. DQ-VAE [18] further encodes images with variable-length codes.

Since quantization is equivalent to tokenizing images, many methods attempt to enable multimodal generation. The dVAE proposed by DALL-E [40] relaxes the discrete sampling problem utilizing Gumbel-Softmax [33, 22], outputting the probability distribution of codebook codes. SEED [11] designed a Causal Q-Former to extract image embeddings and quantize them. LQAE [28] trains VQ-VAE [35] to quantize the image into the frozen LLM codebook space directly. SPAE [56] introduces multi-layer and coarse-to-fine pyramid quantization and semantic guidance with CLIP

[39]. V2L [61] further proposes global and local quantizers. Given the absence of a pre-trained model tailored for IDS, we directly concatenate visual tokens with IDS tokens to performed autoregression.

# 3 Method

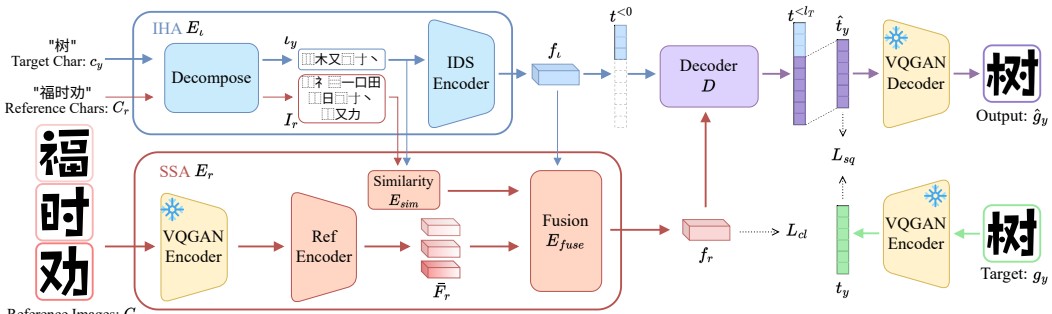

Figure 2: Overview of our proposed method. The overall framework mainly consists of three parts: IDS Hierarchical Analysis module $E_\iota$, Structure-Style Aggregation block $E_r$, and a decoder $D$.

As shown in Fig. 2, given a target character $c_y$, reference characters $C_r = \{c_r^i\}_{i=1}^k$ and reference glyphs $G_r = \{g_r^i\}_{i=1}^k$, the goal of our framework is to generate a glyph $\hat{g}_y$ that conforms to the semantics of $c_y$ and has a style consistent with $G_r$. To achieve this objective, we analyze $c_y$ with IHA to derive its associated IDS $\iota_y$, which is then encoded as a semantic feature $f_\iota$. Likewise, we can obtain the IDS $I_r = \{\iota_r^i\}_{i=1}^k$ corresponding to $C_r$. Following this, we employ the similarity module $E_{sim}$ to assess the relationship between $\iota_y$ and $I_r$. Combined with $f_\iota$ and the output of $E_{sim}$, the features $\bar{F}_r$ corresponding to $G_r$ are fused into the final style feature $f_r$ in the SSA block. $\iota_y$ is reshaped as initial tokens $t^{<0}$, which is fed into the decoder $D$ along with $f_r$ for autoregressive modeling. Finally, the predicted glyph tokens $\hat{t}_y$ are decoded with the pre-trained VQGAN to obtain the generated glyph $\hat{g}_y$.

## 3.1 IDS Hierarchical Analysis

A simple alternative to using a source glyph as input is to directly employ the character itself to control the semantics of the output. However, considering the vast number of characters in Chinese, this approach proves to be impractical due to its expensive cost. Moreover, it overlooks the structural intricacies of characters.

Ideograph Description Sequence is a structural description grammar for Chinese characters defined by the Unicode Standard, which consists of description characters and basic components (mainly Chinese characters) through a prefix notation. Decomposing Chinese characters into their corresponding IDSs can notably streamline the vocabulary, allowing characters with similar structures or components to share common features.

However, a Chinese character may have multiple equivalent IDSs. Many Chinese characters have a top-bottom or left-right structure, the IDCs follows a long-tail distribution, presenting challenges for model training. Fortunately, the left-middle-right structure of Chinese characters can be equivalently represented by two left-right structures. Similarly, the top-middle-bottom structure equals two top-bottom structures. The examples can be found in Fig. 3.

Based on the above observation, we employ a IDS Hierarchical Analysis (IHA) module. Instead of rigidly querying the decomposition table when determining the IDS of a character, we examine whether the character follows a left-middle-right or top-middle-bottom structure. Subsequently, we construct multiple equivalent IDSs for the same character through random selection. To summarize, $c_y$ and $C_r$ are initially decomposed into $\iota_y$ and $I_r$ respectively. In the IDS encoder, $\iota_y$ is padded to the maximum sequence length $l_I$ and encoded into the associated semantic feature $f_\iota \in \mathbb{R}^{l \times c}$.

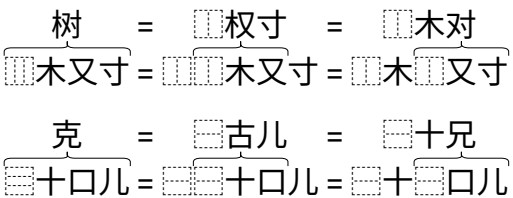

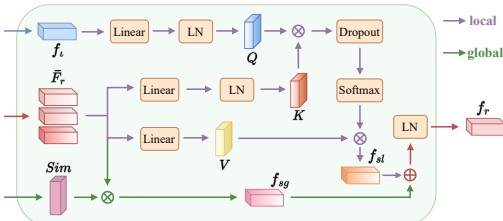

Figure 3: The illustration of equivalent IDSs.

Figure 4: The illustration of the fusion module $E_{fuse}$ of SSA block.

## 3.2 Structure-Style Aggregation

Many previous methods [55, 10, 59, 25, 30, 54] overlook interactions between reference and target characters during the extraction of reference styles, resulting in a lack of relevance in the extracted features. The more closely the reference character resembles the target character, the more effortlessly the generation process can preserve the style. Ideally, employing the target glyph itself as the reference, known as self-reconstruction, should yield the most effective output. Although FS-Font [47] endeavors to ensure that the reference characters cover all components of the target character, its implementation hinges on predefined content-reference mapping, which may limit its adaptability. To address this issue, we propose a Structure-Style Aggregation (SSA) block, as shown in Fig. 2. We convert the reference glyph $G_r$ into the latent space of VQGAN and encode it one by one into the corresponding intermediate features $\bar{F}_r = \{\bar{f}_r^i \in \mathbb{R}^{h \cdot w \times c}\}_{i=1}^k$. The similarity module $E_{sim}$ evaluates the resemblance between each reference IDS $I_r$ and the target IDS $\iota_y$, considering whether they share identical structures or components. It produces a set of similarity weights $Sim = \{sim^i \in \mathbb{R}^1\}_{i=1}^k$, which can guide the subsequent feature fusion process. The fusion module $E_{fuse}$ consists of two branches: global and local style feature aggregation, as shown in Fig. 4. The global features mainly focus on the glyph layout, stroke thickness, and inclination, which can be obtained by merging the coarse style features $\bar{F}_r$ with the similarity weight $Sim$ obtained in the previous step:

$$f_{rg} = \text{softmax}(Sim)\, \bar{F}_r \in \mathbb{R}^{h \cdot w \times c}. \tag{1}$$

While local features are more concerned with the strokes, such as stroke length, stroke edge, and other nuances, we adopt cross-attention to gather the required style feature according to the needs of the target character:

$$
\begin{aligned}
F_s' &= \text{flatten}^2(\bar{F}_r) \in \mathbb{R}^{k \cdot h \cdot w \times c}, \quad Q = \text{LayerNorm}(\text{L}_q(f_\iota)) \in \mathbb{R}^{l \times c}, \\
K &= \text{LayerNorm}(\text{L}_k(F_s')) \in \mathbb{R}^{k \cdot h \cdot w \times c}, \quad V = \text{L}_v(F_s') \in \mathbb{R}^{k \cdot h \cdot w \times c}, \\
A &= \text{dropout}(\frac{QK^T}{\sqrt{c}}) \in \mathbb{R}^{l \times k \cdot h \cdot w}, \quad f_{rl} = \text{softmax}(A)V \in \mathbb{R}^{l \times c},
\end{aligned}
\tag{2}
$$

where $\text{flatten}^2(\cdot)$ denotes flattening the first two dimensions of the feature, and $\text{L}_q, \text{L}_k, \text{L}_v$ are linear projections, and $\text{LayerNorm}(\cdot)$ denotes layer normalization. In Eq. 3, we obtain the aggregated style feature based on Eq. 1 and Eq. 2, where $\circ$ denotes concatenation operator:

$$f_r = \text{LayerNorm}\left(f_{rg} \circ f_{rl}\right). \tag{3}$$

## 3.3 Style Contrast Enhancement

There are some strategies to maintain style consistency: integrating consistency loss [59, 25], introducing a discriminator to determine the generated style [25, 47, 36], or treating the extracted style feature as a variable for further optimization [50]. These approaches are indeed beneficial for improving the generation quality, but they may be inflexible or introduce additional parameters.

In this paper, we propose a streamlined approach named the Style Contrast Enhancement (SCE) module, which promotes the proximity of representations for the same style and the distance between representations for different styles. We apply a linear projection to the style feature $f_r$, resulting in a contrastive feature $e = MLP(f_r)$.

In one batch, we denote the indices of contrastive features corresponding to all samples as $E_* = \{i \in \mathbb{N} \mid 0 \le i < 2N\}$, where $N$ represents the batch size. The dimensionality of $E_*$ is double the

batch size $N$ due to our utilization of a momentum encoder [16]. Each sample $x_a$ within the batch undergoes processing by both the encoder and the momentum encoder, yielding two outputs that serve as positive pairs. The negative sample set is defined as $E_- = \{i \in E_* \mid s(x_i) \neq s(x_a)\}$, while the positive sample set is $E_+ = \{i \in E_* \mid i \neq a, s(x_i) = s(x_a)\}$, where $s(\cdot)$ denotes the operator used to retrieve the corresponding style. The contrastive loss can be calculated as follows:

$$\mathcal{L}_{cl} = -\frac{1}{2N} \sum_{a \in E_*} \log \frac{\sum_{p \in E_+} \exp(e_a^T e_p / \tau)}{\sum_{p \in E_+} \exp(e_a^T e_p / \tau) + \sum_{n \in E_-} \exp(e_a^T e_n / \tau)}. \tag{4}$$

### 3.4 Generation

The decoder $D$ is provided with both semantic feature $f_\iota$ and style feature $f_r$. It treats $f_\iota$ as the initial tokens $t^{<0} = f_\iota$, and then predicts the distribution of the next token autoregressively as $p(t^i \mid t^{<i}, f_r)$. Each newly predicted token is appended to the previous tokens for the subsequent iteration, it continues until all tokens are predicted. The likelihood of the entire sequence can be calculated as $\prod_{i=0}^{l_T-1} p(t^i \mid t^{<i}, f_r)$. There are two ways for incorporating $f_r$. The most straightforward approach involves using $f_r$ as initial tokens, represented by $t^{<0} = f_\iota \circ f_r$, akin to $f_\iota$. These tokens participate in each forward pass, relying on the self-attention mechanism to extract and integrate features.

However, it is only practical for low-resolution scenarios. Viewing $f_r$ as tokens may lead to excessively long sequences, requiring a balance between computational efficiency and generation quality. To address this challenge, we incorporate $f_r$ into each block of the decoder $D$ via cross-attention. The tokens act as queries to align with the corresponding style features. $t^{<l_T}$ denotes all the predicted tokens, from which $t^{<0}$ is removed to get the glyph tokens $\hat{t}_y$. The objective in Eq. 5 is to maximize the log-likelihood of the token sequence.

$$\mathcal{L}_{sq} = -\log\left(\prod_{i=0}^{l_T-1} p(t^i \mid t^{<i}, f_r)\right), \tag{5}$$

Finally, the model can be trained according to the objective in Eq. 6.

$$\mathcal{L}_{total} = \mathcal{L}_{sq} + \lambda_{cl} \mathcal{L}_{cl}, \tag{6}$$

where $\lambda_{cl}$ controls the weight of contrastive loss, cf. Eq. 4, we set $\lambda_{cl} = 0.5$ in our experiments.

## 4 Experiments

### 4.1 Dataset and Evaluation Metrics

**Datasets** We gathered 464 fonts from the Internet, covering diverse categories like printed, handwritten, and artistic styles. Next, we selected 3,500 commonly encountered Chinese characters and rendered them into 128x128 resolution images using the collected fonts.

The training set comprises 3,300 randomly selected Chinese characters and 424 fonts, referred to as Seen Fonts and Seen Characters (**SFSC**). There are two test sets: the first includes the same 3,300 characters but with different 40 fonts, called Unseen Fonts and Seen Characters (**UFSC**). The second test set consists of the remaining 200 characters and the same 40 fonts, known as Unseen Fonts and Unseen Characters (**UFUC**). We found a publicly accessible IDS

Table 1: 12 IDCs used in this paper.

| IDC | Structure | Example (Char:IDS) |
|-----|-----------|--------------------|
| ⿰ | left-right | 鸿:⿰江鸟, 厶:⿰乙丶 |
| ⿱ | top-bottom | 惹:⿱若心, 主:⿱丶王 |
| ⿲ | left-middle-right | 鸿:⿲氵 工鸟, 小:⿲亅丿丶 |
| ⿳ | top-middle-bottom | 惹:⿳艹右心, 叁:⿳厶大三 |
| ⿴ | enclosed-surrounding | 回:⿴口口, 又:⿴又丶 |
| ⿵ | left-top-right-surrounding | 闪:⿵门人, 太:⿵大丶 |
| ⿶ | left-bottom-right-surrounding | 山:⿶凵丨, 义:⿶乂丶 |
| ⿷ | top-left-bottom-surrounding | 匠:⿷匚斤, 兔:⿷⺈兔丶 |
| ⿸ | top-left-surrounding | 友:⿸ナ又, 厌:⿸厂犬 |
| ⿹ | top-right-surrounding | 乃:⿹乃丿, 勺:⿹勹丶 |
| ⿺ | left-bottom-surrounding | 边:⿺辶力, 犬:⿺大丶 |
| ⿻ | overlaying | 平:⿻干丷, 丸:⿻九丶 |

decomposition table[3]. However, it exhibits several redundant entries and circular references, as well as an absence of some characters. Therefore, we performed simplifications and enhancements, reducing the number of IDCs to the 12 depicted in Table 1, which is sufficient for most frequently used Chinese characters. For convenience, we set the basic component's IDS as itself.

---

[3] https://babelstone.co.uk/CJK/IDS.TXT

**Evaluation metrics** We compare all methods in the following metrics, i.e., FID [17], L1, LPIPS [57], RMSE, and SSIM [52]. Since aesthetics is inherently subjective, we conduct a user study for all methods to evaluate model performance based on user satisfaction. We observe that the existing font generation methods have differences in data preprocessing and metric selection. Factors such as glyph resolution, the padding around glyphs, the range of pixel values, the number of reference glyphs, and the evaluation function implementation all influence metric values. For example, NTF [10] and CF-Font [50] center the glyph within the canvas, leaving white space around it. However, this leads to inflated metric calculations. To ensure a fair comparison, we adopt consistent test data and metric implementation across all methods under evaluation. Specifically, we eliminate padding around the glyphs, fix the canvas resolution to 128 pixels, scale the data range to $[0, 1]$, utilize SqueezeNet [20] as the network type to calculate LPIPS [57], and select the inceptionv3 [46] feature layer with 2048 dimensions for FID [17] calculation.

Table 2: Quantitative evaluation on UFSC and UFUC dataset. "User" indicates user study, the samples are generated under 3-shot setting. Bold and underlined numbers denote the best and the second best respectively. Please refer to Fig. 10 in Appendix for the corresponding radar plots.

| | Methods | 1shot | | | | | 3shot | | | | | 8shot | | | | | User (%)↑ |
|---|---|---|---|---|---|---|---|---|---|---|---|---|---|---|---|---|---|
| | | FID↓ | L1↓ | LPIPS↓ | RMSE↓ | SSIM↑ | FID↓ | L1↓ | LPIPS↓ | RMSE↓ | SSIM↑ | FID↓ | L1↓ | LPIPS↓ | RMSE↓ | SSIM↑ | |
| UFSC | CG-GAN [25] | 11.3911 | 0.1784 | 0.1500 | 0.3997 | 0.4428 | 10.8713 | 0.1771 | 0.1464 | 0.3982 | 0.4441 | 11.1332 | 0.1764 | 0.1457 | 0.3974 | 0.4440 | 14.78 |
| | LF-Font [38] | 32.9264 | 0.1764 | 0.1586 | 0.3967 | 0.4465 | 29.1840 | 0.1786 | 0.1576 | 0.3998 | 0.4432 | 26.9590 | 0.1694 | 0.1567 | 0.3875 | 0.4590 | 11.42 |
| | FS-Font [47] | 112.0971 | 0.2836 | 0.3108 | 0.5145 | 0.2795 | 25.5231 | 0.2075 | 0.1916 | 0.4343 | 0.3865 | 93.7912 | 0.1900 | 0.2086 | 0.4124 | 0.4183 | 6.55 |
| | CF-Font [50] | 20.4457 | 0.1839 | 0.1581 | 0.4066 | 0.4323 | 30.8426 | 0.1767 | 0.1650 | 0.3977 | 0.4468 | 30.9829 | 0.1784 | 0.1595 | 0.3990 | 0.4465 | 12.22 |
| | VQ-Font [36] | 72.7064 | 0.1958 | 0.2215 | 0.4201 | 0.4077 | 32.9390 | 0.1789 | 0.1775 | 0.4016 | 0.4405 | 33.6378 | 0.1774 | 0.1732 | 0.3995 | 0.4413 | 7.50 |
| | NTF [10] | 35.3797 | 0.2602 | 0.2027 | 0.4887 | 0.3244 | 26.1215 | 0.2275 | 0.1749 | 0.4542 | 0.3659 | 23.0519 | 0.2238 | 0.1739 | 0.4501 | 0.3720 | 6.43 |
| | FontDiffuser [55] | **3.9969** | 0.1938 | 0.1371 | 0.4180 | 0.4076 | **3.6989** | 0.1774 | 0.1248 | 0.3980 | 0.4370 | **4.1017** | 0.1748 | 0.1234 | 0.3947 | 0.4420 | 18.32 |
| | IF-Font (Ours) | 6.7695 | **0.1529** | **0.1307** | **0.3688** | **0.4915** | 6.8359 | **0.1478** | **0.1258** | **0.3620** | **0.5021** | 6.7383 | **0.1429** | **0.1216** | **0.3552** | **0.5140** | 22.78 |
| UFUC | CG-GAN [25] | 13.4734 | 0.1805 | 0.1508 | 0.4019 | 0.4362 | 13.0347 | 0.1790 | 0.1471 | 0.4001 | 0.4383 | 13.2049 | 0.1780 | 0.1462 | 0.3991 | 0.4391 | 15.73 |
| | LF-Font [38] | 37.3840 | 0.1835 | 0.1620 | 0.4047 | 0.4283 | 28.8252 | 0.1850 | 0.1609 | 0.4071 | 0.4283 | 30.5147 | 0.1735 | 0.1582 | 0.3920 | 0.4473 | 11.65 |
| | FS-Font [47] | 112.6636 | 0.2847 | 0.3112 | 0.5155 | 0.2764 | 31.2833 | 0.2106 | 0.1923 | 0.4373 | 0.3785 | 98.9486 | 0.1921 | 0.2095 | 0.4146 | 0.4131 | 6.17 |
| | CF-Font [50] | 22.8601 | 0.1865 | 0.1584 | 0.4094 | 0.4259 | 34.0245 | 0.1796 | 0.1660 | 0.4009 | 0.4399 | 33.2477 | 0.1809 | 0.1601 | 0.4019 | 0.4399 | 12.03 |
| | VQ-Font [36] | 75.1737 | 0.1980 | 0.2217 | 0.4223 | 0.4018 | 36.4831 | 0.1809 | 0.1776 | 0.4037 | 0.4345 | 36.5486 | 0.1796 | 0.1733 | 0.4017 | 0.4354 | 8.23 |
| | NTF [10] | 39.3581 | 0.2678 | 0.2074 | 0.4958 | 0.3086 | 29.9205 | 0.2303 | 0.1753 | 0.4568 | 0.3593 | 27.9580 | 0.2290 | 0.1755 | 0.4553 | 0.3619 | 6.33 |
| | FontDiffuser [55] | **8.2524** | 0.1914 | 0.1527 | 0.4157 | 0.4163 | **7.6444** | 0.1771 | 0.1413 | 0.3981 | 0.4418 | **8.9166** | 0.1702 | 0.1367 | 0.3890 | 0.4543 | 18.57 |
| | IF-Font (Ours) | 8.4844 | **0.1651** | **0.1387** | **0.3845** | **0.4676** | 8.4922 | **0.1597** | **0.1338** | **0.3775** | **0.4782** | 8.3203 | **0.1561** | **0.1305** | **0.3728** | **0.4864** | 21.28 |

## 4.2 Comparison with state-of-the-art Methods

We compare the proposed IF-Font with seven SOTA methods on our UFSC and UFUC datasets respectively, including CG-GAN [25](CVPR 2022), LF-Font [38](TPAMI 2022), FS-Font [47](CVPR 2022), CF-Font [50](CVPR 2023), VQ-Font [36](ICCV 2023), NTF [10](CVPR 2023) and FontDiffuser [55] (AAAI 2024). All methods are trained from scratch on our SFSC dataset according to their respective official codes and default configurations. We slightly modify the codes of CG-GAN, LF-Font, FS-Font, VQ-Font and FontDiffuser to support varied numbers of reference glyphs.

### 4.2.1 Quantitative comparison

Table 2 compares IF-Font and other SOTA methods. IF-Font significantly surpasses competitors in all reference glyph number settings for both datasets. Notably, IF-Font's performance on FID metric is exceptionally low, reaching a single-digit score, thanks to the high quality and clarity of the samples it generates. FS-Font [47] relies heavily on the predefined content-reference mapping, whereas the reference glyphs in all our experiments are randomly selected. Especially, when only one reference glyph is provided, covering all components of the target character becomes challenging, leading to poor performance of FS-Font, as shown in Fig. 10a. NTF [10] also struggles to imitate the target style, the layout of its generated samples often resembles that of the source font. In cases where there's a significant disparity between the source and target styles, NTF is prone to missing strokes.

We attribute CF-Font's performance to its reliance on fusing contents of 10 basic fonts. However, there happen to be a gap between the train dataset and our evaluation dataset. We conduct a user study through Fuxi Youling Crowdsourcing Platform [4] to quantify the subjective quality. For each test dataset, 5 characters are randomly selected, and each model is required to generate glyphs corresponding to 40 unseen fonts. A total of 30 participants are asked to select the option that most closely matches the ground truth from the generated results. The outcomes of the user study are presented in the last column of Table 2.

---

[4] https://fuxi.163.com/solution/data

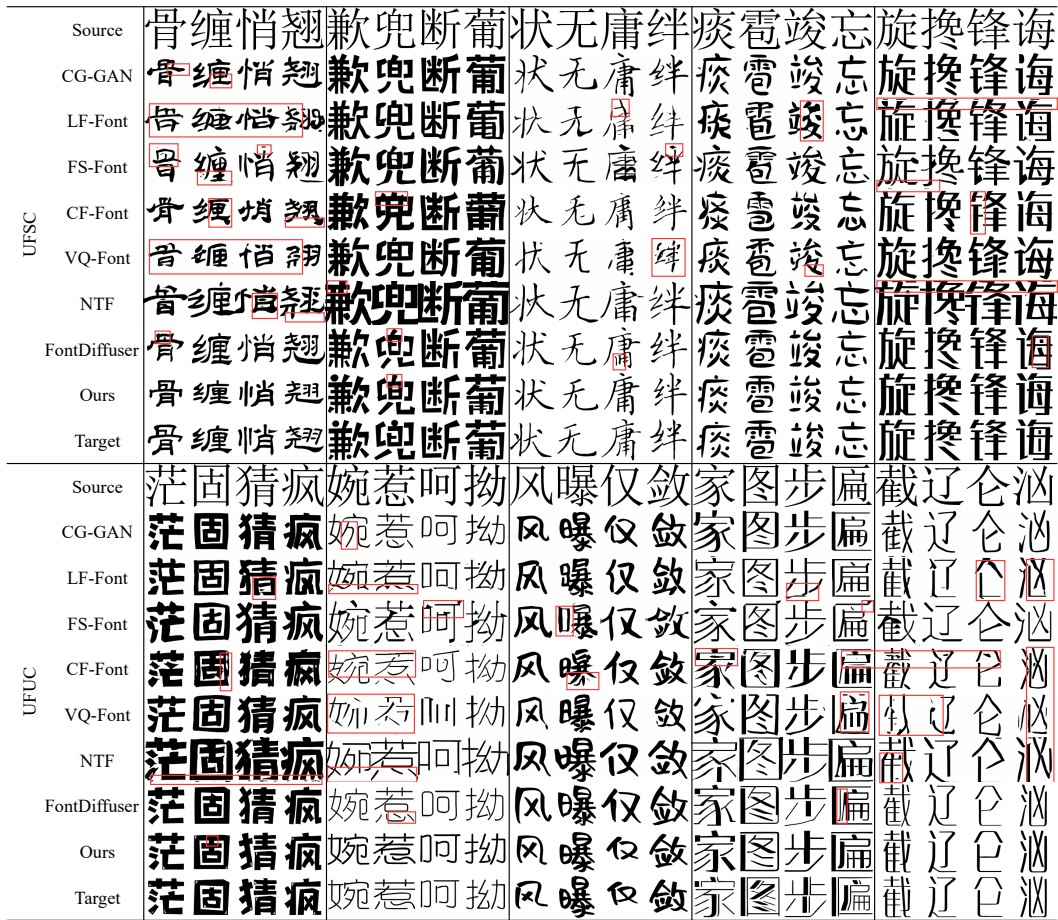

Figure 5: Qualitative comparison with state-of-the-art methods, in which red boxes outline the artifacts. "Source" denotes the content glyph of other methods, IF-Font only employs the corresponding IDS.

#### 4.2.2 Qualitative comparison

We present the corresponding samples from Table 2 in Fig. 5. IF-Font stands out by producing the clearest and most style-consistent samples. In contrast, FS-Font [47], LF-Font [38], CF-Font [50], and other models exhibit issues such as stroke errors or blur. VQ-Font [36] and NTF [10] are constrained by the source font and struggle with flat or narrow layouts, resulting in incorrect structures. VQ-Font even tends to crop marginal parts of glyphs to fit the target style. While CF-Font generally preserves the correct glyph layout, its outputs exhibit noticeable artifacts, indicating some remaining style inconsistencies. The performance of FontDiffuser [55] is also outstanding, but there is still a slight deficiency in the imitation of font styles. On the other hand, IF-Font maintains the correct character structures and excels in aspects such as the aspect ratio, glyph layouts, and stroke details.

Table 3: Ablation studies on different modules. The first row is the results of baseline. I, S and C represent IHA, SSA, and SCE respectively.

| Module | | | FID↓ | L1↓ | LPIPS↓ | RMSE↓ | SSIM↑ |
|---|---|---|---|---|---|---|---|
| I | S | C | | | | | |
| ✗ | ✗ | ✗ | 8.2656 | 0.1632 | 0.1383 | 0.3820 | 0.4728 |
| ✓ | ✗ | ✗ | 8.3750 | 0.1638 | 0.1381 | 0.3828 | 0.4764 |
| ✓ | ✓ | ✗ | **7.5391** | 0.1614 | 0.1348 | 0.3797 | 0.4780 |
| ✓ | ✓ | ✓ | 8.4922 | **0.1597** | **0.1338** | **0.3775** | **0.4782** |

| IDS | Reference | Baseline | +I | +IS | +ISC | Target |
|---|---|---|---|---|---|---|
| ⿱⿴⿰示⿰几乂 | 福 | 飘 | 飘 | 飘 | 飘 | 飘 |
| ⿲木木彡 | 福 | 彬 | 彬 | 彬 | 彬 | 彬 |
| ⿲厶大三 | 福 | 叁 | 叁 | 叁 | 叁 | 叁 |
| ⿰氵⿱口丁一鸟 | 福 | 鸿 | 鸿 | 鸿 | 鸿 | 鸿 |

Figure 6: Visualization of different modules in Table 3. Red, blue and green boxes represent the missing components, style inconsistency and corresponding improvements respectively.

### 4.3 Ablation Studies

**Main modules**   Removing the IHA, SSA, and SCE modules of IF-Font, a baseline model can be obtained. For a input character, it directly looks up the decomposition table to derive the corresponding IDS, and then encode the semantic feature $f_\iota$ through a embedding layer. The intermediate style features $\bar{F}_r$ are directly averaged as style features $f_r$, excluding any interactions with the similarity weight $Sim$ and semantic feature $f_\iota$. The whole model relies solely on cross-entropy loss for supervision. Building upon the baseline, we incrementally reintegrate three modules to assess their individual contribution. Quantitative results are presented in Table 3, while Fig. 6 provides visualizations of these results. For further ablation study of the SSA block, please refer to Table 6 in Appendix B.1. Upon integrating our modules, a consistent improvement is observed across most metrics. Fig. 6 illustrates how IHA alleviates the issue of missing components present in the baseline. SSA enhances style consistency, while SCE improves the capability to imitate styles.

Table 4: The impact of IDS granularity on performance.

| Granularity | FID↓ | L1↓ | LPIPS↓ | RMSE↓ | SSIM↑ |
|---|---|---|---|---|---|
| Component | 8.4922 | **0.1597** | **0.1338** | **0.3775** | 0.4782 |
| Stroke | **8.4297** | 0.1616 | 0.1347 | 0.3799 | **0.4888** |
| Mixed | 8.5234 | 0.1598 | 0.1337 | 0.3775 | 0.4782 |

**IDS granularity**   We further analyze the impact of three different IDS granularities: components, strokes and mixed. Please see Table 5 in the Appendix for the examples of these granularities. Table 4 shows the quantitative results. Stroke granularity results in performance degradation across three metrics. We attribute this decline to the conflict between IDSs, hindering the model's ability to identify the target character. An attempt to concatenate IDSs from both granularities yields performance akin to that of component granularity. While this approach extends the sequence considerably, hence we opt for component granularity.

### 4.4 Visualization of SSA

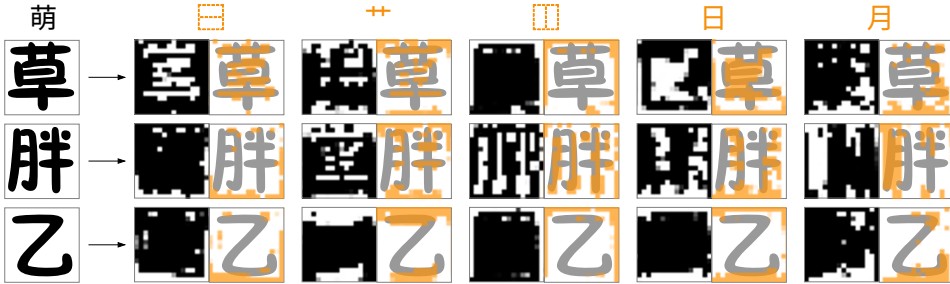

Figure 7: Visualization of attention maps between IDS and reference glyphs. The symbols above are the target character (black) and the corresponding IDS (orange).

To demonstrate the effectiveness of Structure-Style Aggregation block, we visualize the attention maps corresponding to each IDC and component within the IDSs relative to the reference glyphs, as depicted in Fig. 7. Specifically, we choose the matrix $A \in \mathbb{R}^{l \times k \cdot h \cdot w}$ in local feature calculation. For each position $i$ of the target IDS$\iota_y$, there exists a corresponding attention map $A^i \in \mathbb{R}^{k \cdot h \cdot w}$, which indicates the attention that $\iota_y^i$ pays to the $k$ reference features. We present the attention map $A^i$ to visualize the distribution of attention weights directly. Additionally, we apply opacity to this map and overlay it onto the original reference glyph.

As we can see, when the target IDC or component exists in the reference glyph, more attention will be paid to the corresponding place. For instance, in the first row, the first, second, and fourth columns, and in the second row, the third and fifth columns are distinctly highlighted. Conversely, if the reference glyph lacks the target component, the local branch tends not to engage, as evidenced by the nearly blank third row. This approach stems from a preference to avoid forced attention allocation which might lead to interference. Instead, leveraging the average style captured by the global branch helps maintain a baseline quality of the output.

## 4.5 New Glyph Creation

We validate the flexibility of IF-Font by generating the glyphs using IDSs of non-standard Chinese characters. Fig. 8 shows our experimental results. IF-Font demonstrates robust generalization by following given IDSs to produce new glyphs with accurate structure and consistent style. We fixed the font to "Sarasa Gothic" in the experiment, which is a CJK programming font. The last two columns lack ground truths due to their entirely non-existent characters.

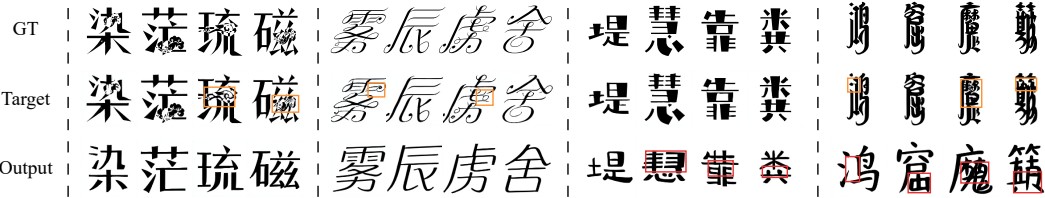

Figure 8: The ability of the IF-Font to create glyphs. The first two columns are kokuji, and the last two columns are completely non-existent Chinese characters.

## 5 Discussion

Figure 9: Failed cases on complex fonts of UFUC. Orange boxes highlight reconstruction errors of VQGAN, red outlines the structural errors. *GT*: the glyphs rendered by fonts; *Target*: the glyphs reconstructed by VQGAN; *Output*: the glyphs generated by IF-Font.

**Failure cases** Although our method enables high-quality generation under most circumstances, it still struggles on some hard cases, as illustrated in Fig. 9. IF-Font encounters difficulties with fancy and irregular font styles, including those with decorations, extremely flat or narrow layouts, excessively curved strokes, and calligraphic writings. Despite these challenges, it continues to preserve the correct character structure. Further discussion on the reasons for the difficulties in generating these fonts can be found in Appendix C.3.

**Usability** We focus on CJK characters due to their unique spatial structures, which better reflect the characteristics of our method. By expanding the vocabulary and incorporating relevant data for training, IF-Font can also be adapted to handle other character sets.

**Advantages** *Conforms to writing habits*. We believe that the process of autoregressive modeling with IDS implicitly contains the order of writing. *Scalability*. Good scalability can be achieved by leveraging the mature experience of LLMs. *Robustness*. Due to vector quantization, glyphs are represented by a limited number of tokens (only 256 types), which reduces the learning difficulty for the decoder and decreases the likelihood of artifacts and other issues in the generated results.

## 6 Conclusion

We have presented IF-Font, a novel font generation paradigm. IF-Font redefines font generation as a sequence prediction task by quantizing glyphs as token sequences and leveraging Ideographic Description Sequence (IDS) to control the semantics of the generated glyphs. This method demonstrates exceptional capability in managing complex styles while preserving the correct structures. Benefiting from the flexibility of IDS, our method enables the creation of glyphs. This is achieved by formulating legal IDSs, which is a salient advantage over other methods that typically require the character to be present in at least one font as a precondition for generation. Refining and improving the IDS decomposition rules is considered future works. Furthermore, exploring the integration of IDS into handwritten font generation may yield interesting insights.

## Acknowledgments and Disclosure of Funding

This work was supported in part by the National Key Research and Development Plan of China under Grant 2021YFB3600503, in part by the National Natural Science Foundation of China under Grant 61972097 and U21A20472, in part by the Major Scientific Research Project for Technology Promotes Police under Grant 2024YZ040001, in part by the Natural Science Foundation of Fujian Province under Grant 2021J01612 and 2020J01494.

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

# A  Experiment Details

## A.1  Implementation Details

IF-Font was trained in a server equipped with an Intel Xeon Silver 4110 CPU, 128 GB of RAM, and an NVIDIA Tesla V100 PCIe 16GB GPU. The training takes about 42 hours for 15 epochs with batch size of 128.

The VQGAN model in IF-Font loads pre-trained weights. It has a codebook size of 256, and the encoder downsamples the image with a factor of 8. Consequently, the length of the codebook indices corresponding to a single glyph is $l_T = 256$, whereas the IDS have a fixed length of $l_I = 35$. Our decoder consists of 10 Transformer blocks, each integrating a self-attention layer, a cross-attention layer, and a multi-layer perceptron (MLP). We have configured the model with 8 attention heads and a feature dimension of 384. In IF-Font, parameters are optimized using the AdamW optimizer [32], which employs a learning rate schedule that includes warmup and cosine annealing.

## A.2  Training Data

We primarily obtain fonts from the Foundertype platform under a personal non-commercial academic research license. We present the examples of three IDS granularities in Table 5.

Table 5: Different IDS granularities. Mixed means that the IDSs of both granularities are used.

| Granularity | IDS | Length |
|---|---|---|
| Componet | 囗犭 日艹田 | 5 |
| Stroke | 囗口㇆丶 丿丿 日㇀一囗丨 丨囗日囗丨 ㇇一㇀一丨 | 21 |
| Mixed | 囗犭 日艹田[sep]囗口㇆丶 丿丿 日㇀一囗丨 丨囗日囗丨 ㇇一㇀一丨 | 27 |

# B  Additional Results

## B.1  Further ablation of SSA

Table 6: Ablation studies on both branches of SSA.

| SSA | FID↓ | L1↓ | LPIPS↓ | RMSE↓ | SSIM↑ |
|---|---|---|---|---|---|
| w/o global | **7.9766** | 0.1607 | 0.1347 | 0.3789 | 0.4775 |
| w/o local | 8.2578 | 0.1620 | 0.1364 | 0.3805 | 0.4756 |
| full | 8.4922 | **0.1597** | **0.1338** | **0.3775** | **0.4782** |

## B.2  Radar plots

Fig. 10a depicts the one-shot setting, where style extraction poses a considerable challenge, leading to comparable performance across most models. Nevertheless, as the number of reference glyphs increases, the advantage of IF-Font becomes progressively apparent in Fig. 10b and 10c.

# C  Further discussion

## C.1  Quantization accuracy

Since the target glyph is reconstructed from a quantized sequence, the accuracy of this reconstruction imposes a ceiling on IF-Font's potential performance. As shown in Figure 9, there are slight differences between the target and the ground truth. Given that the glyphs are grayscale images and

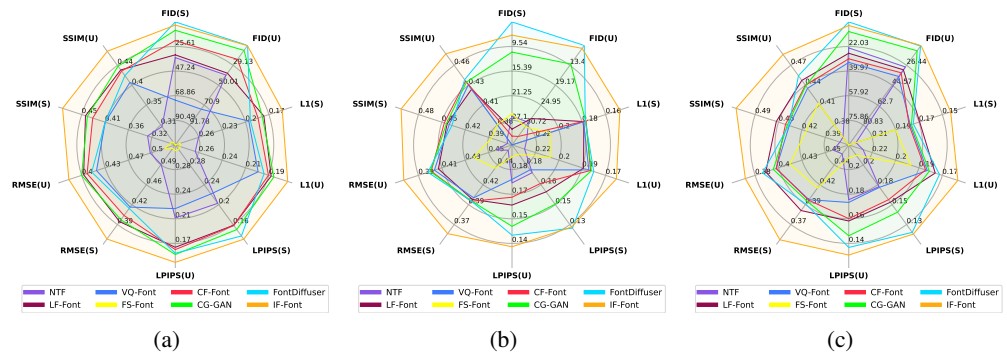

(a)        (b)        (c)

Figure 10: Compared with the methods based on the content-style disentanglement paradigm, IF-Font achieves state-of-the-art performance on all metrics under three few-shot settings. The metrics are annotated with brackets in the figure to specify the dataset used for evaluation: (S) represents UFSC, and (U) refers to UFUC. (a) 1-shot setting. (b) 3-shot setting. (c) 8-shot setting.

relatively simple, fine-tuning the VQ-GAN or switching to a superior quantized method is expected to minimize the loss of accuracy. We use original VQ-GAN checkpoints in order to highlight our main contributions.

## C.2 The shortcomings of IDS

In fact, IDS is still not sufficiently perfect to identify Chinese characters. On the one hand, there are rule conflicts: a few Chinese characters are too similar, and their IDSs of stroke granularity are exactly the same. On the other hand, the spatial descriptions are insufficiently clear. For example, the left-right structure indicates that two components are placed on the left and right, but the distance between them is not specified, which requires the model to distinguish them through sufficient learning.

## C.3 Complete results on novel fonts

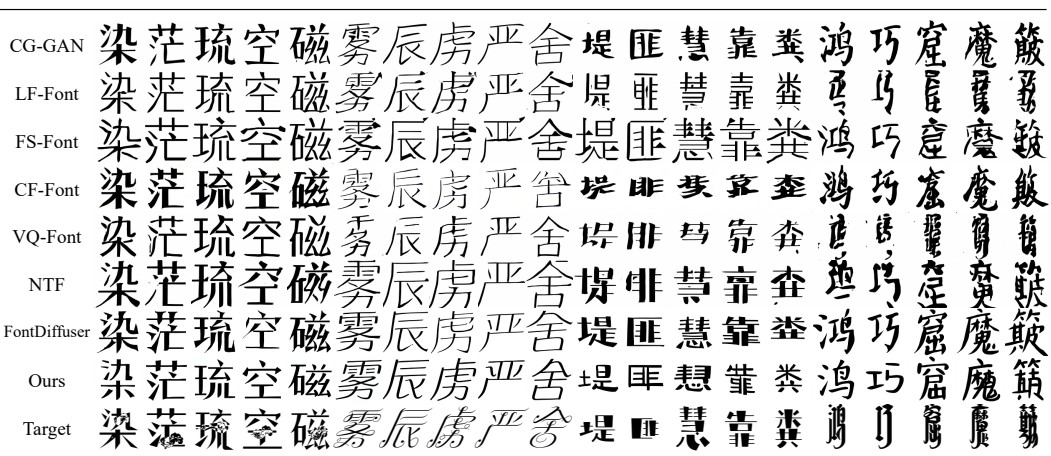

Figure 11: Complete results generated by methods with novel fonts of Fig. 9.

We believe that generating novel fonts presents significant challenges for several reasons. First, they substantially deviate from standard character structures, involving a trade-off between the number of references and the quality of generation. Furthermore, font design involves subjectivity and randomness. For instance, in the first font depicted in Fig. 9, the position, size, and shape of the auspicious cloud patterns are the result of manual design. It is important to note that this issue is common across all font generation methods. Similar discussions can be found in LF-Font [38] and

CF-Font [50]. Unfortunately, this limitation has not been well addressed yet. We demonstrate the results of other methods on novel fonts in Fig. 11 to support the above conclusion.

## C.4 Restrictions

To achieve high-quality generation, IF-Font requires training data to cover as many character components and font styles as possible. Due to the utilization of attention mechanisms, IF-Font entails quadratic computational complexity related to sequence length, while inference relies on autoregressive processes, resulting in a slow sampling speed.

## D Details of the user study

To be accurate, we pay the Fuxi Youling Crowdsourcing Platform (a third-party platform) to conduct the user study. The participants in the user study are users of that platform, who will be informed of the full details in advance. We limit the number of participants to 30, and they are allowed to choose whether or not to participate and can complete the evaluation at any time and in any location using their own devices (usually a phone or computer).

All model names in the evaluation are replaced with numbers, and participants only need to select the option that best matches the ground truth from the model results displayed on the page. After the evaluation, the platform pays participants and provides us with de-identified model effectiveness data. We have no direct contact with participants and are unable to obtain their specific identities.

## E Broader Impact

IF-Font could help to improve the productivity and creativity of font designers, and there is also hope for preserving ancient calligraphic works. In addition, IF-Font is capable of imitating the font style from as little as one reference glyph. That makes it easy to reproduce commercial font designs, raising concerns regarding potential copyright infringement. To safeguard the rights of font creators, we urge users to adhere to license requirements and call for the responsible use of our method.

## F Licenses

We present a complete list of references and licenses in Table 7 for all the existing assets we used in this work.

Table 7: License information for the existing assets used.

| Software Code | URL | License |
|---|---|---|
| VQGAN | Link | MIT license |
| nanoGPT | Link | MIT license |
| CG-GAN | Link | N/A |
| LF-Font | Link | MIT license |
| FS-Font | Link | MIT license |
| CF-Font | Link | N/A |
| VQ-Font | Link | N/A |
| NTF | Link | MIT license |
| torchmetrics | Link | Apache v2.0 |

