# OpenReview forum: "IF-Font: Ideographic Description Sequence-Following Font Generation"
_NeurIPS.cc/2024/Conference — NeurIPS 2024 poster_

### Official Review · Reviewer_o7CM · 2024-06-14

**Soundness:** 3
**Presentation:** 3
**Contribution:** 3
**Rating:** 5
**Confidence:** 4

**Summary:**

This paper presents a method that generate Chinese glyphs using token prediction approach. The core contribution is to leverage the concept called Ideographic Description Sequence and develop a network architecture to generate IDS that represent the target character.

**Strengths:**

The strengths of this paper are:
- The problem formulation using Ideographic Description Sequence is novel and interesting.
- The evaluation is thorough and the generated Chinese glyphs are convincing.

**Weaknesses:**

The weaknesses of this paper are:
- The exposition is sometimes confusing. I would encourage the authors to explicit show the output of the network before putting the output together as a glyph (I suppose the output sequence is composed of multiple tokens.)
- I understand the proposed method aims to focus on generating Chinese characters, but it seems can not be generlized to other writing systems, including Roman characters?

**Questions:**

My questions and concerns are:
- The exposition of the paper is quite unclear to me. In the paper, I did not find an explanation of what each token really represents. Is it a part of the glyph? The decoded results shown in the paper are always a completed glyph.
- It is unclear how to compose the predicted token into a completed glyph.
- I recommend the authors to survey and discuss more about Ideographic in previous font manipulation is not thorough.
    - e.g., Ariel Shamir, A. Rappoport, Compacting oriental fonts by optimizing parametric elements, The Visual Computer, 1999
- It is unclear what does "IDS is not perfect enough to identify Chinese characters" in the limitation section mean?

**Limitations:**

- as mentioned above, I am curious how well the method can generalized to other writing system? If it is hard, then I suggest authors discuss this in the limitation section.

---

> ### Author Rebuttal · Authors · 2024-08-07
>
> ### Q1: what is the output of the network before putting the output together as a glyph? Is it composed of multiple tokens?
> Yes, the output sequence consists of multiple tokens. Please refer to the right of Figure 1, where the small green squares around Transformer Decoder are tokens (vector quantized tokens), which are **the indexes of codes in the VQ-GAN codebook** (VQ-VAE or other vector quantized methods are also applicable).
>
> Specifically, after encoding an image into a feature map, each feature vector is replaced by the closest code in the codebook. Since all vectors are selected from the codebook, we can use the corresponding indexes to replace them, yielding a 2D array of integers. This array is then flattened into a sequence where each integer represents a token.
>
> ### Q2: How to compose the predicted token into a completed glyph?
> The token sequence is first restored to a 2D array and replaced by vectors from the codebook, resulting in a quantized feature map. This feature map is then **decoded by a VQ-GAN decoder** to restore the original image. Since our work focuses on modeling the tokens of glyphs, decoding the predicted tokens into a glyph falls beyond the paper’s scope
>
> ### Q3: The survey and discussion about Ideographic in previous font manipulation is not thorough.
> Thank you for providing a valuable reference, which we carefully read and found that it deals with the parametric representation of glyphs, allowing for a elegant trade-off between glyph quality and the amount of compression. It seems to be more closely related to the VQ-GAN we used, both trying to compress glyph representations. In future work, using this parametric elements as an input/output format may be promising.
>
> Following your suggestion, we will include relevant discussions and cite this reference in the final submission. Thanks for your kind response.
>
> ### Q4: what does "IDS is not perfect enough to identify Chinese characters" in the limitation section mean?
> The shortcomings of IDS are reflected in two aspects:
> 1. **Rule conflicts**. A very small number of characters are too similar, and their IDS of stroke granularity is identical.
> 2. **Insufficiently clear spatial descriptions**. For example, the left-right structure puts two components on the left and right, but the distance between them is not precisely specified, which requires the model to learn enough to distinguish.
>
> ### Q5: How well the method can generalize to other writing systems?
> We focused on CJK characters because they have spatial structures, can be used to better demonstrate our main contributions. We have not tested it on other writing systems yet and are uncertain about its generalizability. Thanks for your valuable feedback, we will clearly state this in the manuscript.

---

> > ### Comment · Reviewer_o7CM · 2024-08-10
> >
> > Thank you for your thorough response and I think most of my concerns are addressed.

---

### Official Review · Reviewer_ufc8 · 2024-06-16

**Soundness:** 2
**Presentation:** 3
**Contribution:** 4
**Rating:** 7
**Confidence:** 3

**Summary:**

This paper approaches the task of Few-shot Font Generation (FFG) for Chinese characters, proposing to model the target glyph with Ideographic Description Sequence (IDS) tokens to achieve style-content disentanglement. Reference font images and the target character’s IDS are fed into a VQGAN-based pipeline to decode the output image autoregressively as quantized codebook tokens. The results show that this method outperforms SOTA in one- and few-shot settings and for significantly differing font styles.

**Strengths:**

The paper approaches an important problem and the idea of using IDS as input for FFG is clever and logical. The qualitative and quantitative results of the essential idea are convincing and the array of comparisons and attention to fair evaluation is appreciated. The presentation is mostly clear.

**Weaknesses:**

The differences in most metrics when ablating components (Table 3) mostly seem quite small, compared to the larger differences in metrics when comparing to competing methods (Table 2). It’s not immediately convincing that the added complexity of these components (particularly IHA) are justified. On the other hand, the significant drop in FID when using I+S without C in Table 2 seems important and is not explained. If the use of these components gives a qualitative advantage that is not fully reflected in the metrics, this should be illustrated or included in the user study.

The design of the SSA (Sec 3.2) includes sub-components (global and local branches) which are not ablated. It would also help to indicate which is which in Figure 4. Claims about these controlling features such as stroke width, edges etc. (L134-147) could be justified. I also found Sec 4.4 hard to follow; how does Figure 7 show that SSA is effective? It’s not obvious where the attention should be focussed for positioning IDC’s or for those that don’t appear in the reference glyphs.

The method is tested on data collected by the authors. It is unclear if the proposed method would outperform the competing methods (Sec 4.2) if trained on datasets used in prior works.

**Questions:**

Is this approach specific to CJK characters? If so, it seems like this should be mentioned in the title, abstract, and conclusion.

The use of “SCA” is confusing as in L248 it is mentioned as a “module”, but the acronym “SCA” does not appear before. Assuming this refers to “Style Contrast Augmentation” (Sec 3.3) this seems to be the contrastive learning loss term and not a module in the model architecture. I’m also not sure if “augmentation” is the right term since this is not performing data augmentation.

What does it mean on L111 that directly employing the character would be “impractical”? Isn’t this done in prior works? Additionally, L119 says that the long-tail distribution of IDC’s poses a challenge for training, but isn’t this shown in the first row of Table 3 with overall good results? (relative to the competing methods in Table 2)

Will the curated data and IDS table (L202) be made available?

The paper mentions a user study (L208 etc.). Who were the participants, were they paid (L236 mentions “volunteers”), and does this require IRB approval?

Typos: L19-20 incomplete sentence, L199 L131-136 L217 L311 L325-326 grammar, L139 L176 L250 formatting issues, L497 missing period.

**Limitations:**

Limitations and societal implications are adequately discussed.

---

> ### Author Rebuttal · Authors · 2024-08-07
>
> ### Q1: The differences in most metrics seem quite small
> On one hand, we attribute this to **the advantages brought by the new paradigm**. The usage of IDSs and VQ-token based decoder have already achieved good results, making our baseline strong.
>
> On the other hand, this is due to **marginal effects of the metrics**. Because the metrics of baseline are already quite excellent, the improvement brought by modules is relatively weak in terms of metrics.
>
> We show the visualization in Figure 6, and these changes may not be important for all samples, but they are crucial for certain glyphs and effectively improve generation quality.
>
> ### Q2: The significant drop in FID when using I+S without C
> Because FID has limitations and a certain gap with human's perception. We provide corresponding visualization in Figure 6, which shows that our module can effectively improve the quality of glyph.
>
> ### Q3: The sub-components of the SSA are not ablated
> Thank you for your insightful comments and observations regarding our model. We have added below the ablated results for the two branches of SSA module:
>
> | SSA (3shot, ufuc)        | FID↓ | L1↓ | LPIPS↓ | RMSE↓ | SSIM↑ |
> |------------|---|---|---|---|---|
> | wo/ global | **7.9766**  | 0.1607  | 0.1347  | 0.3789  | 0.4775  |
> | wo/ local  | 8.2578  | 0.1620  | 0.1364  | 0.3805  | 0.4756  |
> | full        | 8.4922  | **0.1597**  | **0.1338**  | **0.3775**  | **0.4782**  |
>
> Figure R2 (in the rebuttal PDF) is our modified version of Figure 4, with the two branches highlighted. If you have any further suggestions, please feel free to continue discussing with us.
>
> ### Q4: How does Figure 7 show that SSA is effective?
> Figure 7 shows the attention maps for the local branch in SSA. When there are target IDCs or components present in the reference glyphs, more attention is paid to the corresponding areas. For example, the first, second, fourth attention maps in the first row and third, fifth attention maps in the second row are highlighted accordingly.
>
> When a reference glyph does not contain any target elements, the local branch tends to pay less attention to it, which is why the third row appears almost blank.
>
> Rather than forcing attention allocation to introduce interference, we think it is better to adopt the overall features extracted by the global branch.
>
> ### Q5: Why not test methods on datasets used in prior works?
> This is a very reasonable concern. However, in the font generation field, due to **copyright protection**, there is no widely recognized open-source dataset. The current mainstream methods, such as FontDiffuser (AAAI 2024), CF-Font (CVPR 2023), VQ-Font (ICCV 2023), LF-Font (TPAMI 2022), are all use private dataset.
>
> Our dataset is large enough and covers a variety of fonts and most commonly used Chinese characters, We believe that the comparison results are relatively fair. If we use other datasets, we are confident that the proposed method can still outperform the competing methods.
>
> ### Q6: Is this approach specific to CJK characters?
> Our methods is not specific to CJK characters. We focus on CJK characters because of their spatial structures, which can better reflect the characteristics of our method. If we expand the vocabulary and train with corresponding data, IF-Font can also be suitable for other characters.
>
> ### Q7: The use of “SCA” is confusing
> Sorry for the confusion: We define SCA as a module because there are some network layers used to extract features in addition to the loss part. "SCA" refers to "Style Contrast Augmentation", we apologize for not introducing it in Section 3.3, and thank you for pointing it out.
>
> The term "augmentation" might not be suitable, we intended to emphasize the improvement brought by the contrastive loss. Perhaps the term "enhancement" is a better choice? We will continue to refine the wording and make revisions to the draft, any further discussions are welcome.
>
> ### Q8: Why is it “impractical” to directly employ the character
> Some early font generation works have indeed used characters as input directly, but this approach is only suitable for those with a small vocabulary table, such as the Latin alphabet.
>
> In this paper, we focus on CJK characters. Due to their large number, if we were to use them as input directly, an **enormous vocabulary table** is needed. At the same time, new characters are constantly being added, low-frequency characters may lack font support, leading to **insufficient training data**. Directly employing the characters makes model difficult to generalize.
>
> ### Q9: Why is the problem caused by long-tail distribution not obvious in metrics?
> Most Chinese characters are left-right or top-bottom structures, the improvements made by the IHA module are not significant in terms of metrics. We demonstrate this in the fourth cloumn (the column "+I") of Figure 6.
>
> ### Q10: Will the curated data and IDS table be made available?
> Sure, we will release these data to ensure that our model can be reproduced.
>
> ### Q11: Who were the participants in user study, were they paid (L236 mentions “volunteers”), and does this require IRB approval?
> Our user study was conducted through Fuxi Youling Crowdsourcing Platform[1]. We paid the platform to publish a questionnaire with a limited number of 30 slots. After data collection is completed, we received anonymous responses from the platform. Therefore, the term "volunteers" might not be accurate, maybe "platform users" or "participants" would be more suitable?
>
> The platform users fulfill a task to earn points, which can be redeemed for money. Since we collaborated with the third-party platform, IRB approval is not required.
>
> [1] https://fuxi.163.com/solution/data
>
> ### Q12: Typos
> Thank you for your detailed reading of our manuscript. We will correct these typos in our final version.

---

> > ### Comment · Reviewer_ufc8 · 2024-08-09
> >
> > Thank you for your thorough and lucid response. I believe this addresses my concerns so I have updated my rating accordingly. I encourage the authors to incorporate these findings and discussion into the final version of paper.

---

> > > ### Author Response · Authors · 2024-08-09
> > >
> > > We sincerely appreciate the reviewer's insightful suggestions and the increased rating. We will incorporate these findings and discussion into the revised paper.

---

### Official Review · Reviewer_D5bL · 2024-07-12

**Soundness:** 3
**Presentation:** 3
**Contribution:** 2
**Rating:** 5
**Confidence:** 5

**Summary:**

This paper proposed IF-Font handles the task of few-shot font generation via a VQ-GAN based framework. Compared to most existing methods that encode content images, IF-Font only encodes Ideographic Description Sequence (IDS) to convey content information of target characters. Experimental results show the proposed method excels in synthesizing glyphs with neat and correct strokes, and enables the creation of new glyphs based on provided IDS.

**Strengths:**

(1) The presentation is good and this paper is easy to follow.

(2) The proposed method can generate visually pleasing glyph images, from Figure 5.

(3) It is an interesting idea to ONLY encode IDS to model the shape of target characters. However, I have concerns about whether this idea is fully substantiated (see Weaknesses).

**Weaknesses:**

(1) The idea of utilizing component/stroke information of glyphs has already been exploited in many existing papers, such as LF-Font and XMP-Font. What is the key difference that distinguishes IF-Font from those methods?

(3) If the key difference is only encoding IDS to convey content information, I wonder if there is an ablation study verifying that this is a better design than encoding both modalities (i.e., component sequence and glyph images) in XMP-Font?  The current Section 4.4 (Ablations Studies) is a bit unclear to me.

(3) The authors mentioned that they construct multiple equivalent IDSs for the same character through random selection. Does it mean during training, a random IDS is selected to feed the IDS encoder if there are multiple equivalents? Can the IDS encoder resolve the ambiguities in representing the shape of a character? My guess is this is somewhat similar to many-to-one mapping so it can be done. Correct me if I am wrong.

**Questions:**

What is the detailed structure of IDS-Encoder? Is it a Transformer to encode a sequence?

**Limitations:**

The authors discussed the limitations in Section 4.6. Regarding the first limitation (fancy and irregular font styles), I believe the discussion could be expanded further. One key reason might be that novel font styles do not always adhere to the typical topology of characters. For example, the second-to-last font in Figure 5 deviates from the norm, challenging the necessity of using IDS (or pre-defined component sequences). I would like to hear the authors' opinions on this.

---

> ### Author Rebuttal · Authors · 2024-08-07
>
> ### Q1: The key difference between IF-Font and other methods that utilize component/stroke information
> We believe the key difference lies in **whether style-content disentangling is performed**. Previous methods use component/stroke information but still relied on separating and combining corresponding content and style features to generate glyphs. Our method uses IDS as a condition to directly generate characters, rather than morphing the content glyph.
>
> ### Q2: Is the key difference lies in only encoding IDS?
> Perhaps reviewer wonders why our method outperforms previous methods with only one input modality. As mentioned above, **the component/stroke information used in previous methods is not equivalent to IDS**. Hence, it is challenging to attribute the key difference to whether only IDS is used.
>
> We are happy to include an ablation study that uses both modalities to address the reviewers' concerns. However, the addition of the new modality leads to a substantial increase in training costs. We respectfully request permission from the reviewer to present the experimental results in the subsequent discussion period.
>
> ### Q3: Multiple equivalents IDSs for single character
> We apologize that our descriptions are not clear and cause confusion. As shown in Figure 2, an input character is first broken down into single IDS, then fed into IDS encoder during training. IDS encoder analyzes the input IDS using some rules, and if an IDS has multiple equivalents, it randomly selects one for encoding.
>
> Strictly speaking, IDS encoder is not responsible for resolving ambiguities. Since IDS is similar to text, there are cases where different descriptions refer to the same object. **We designed IDS encoder to present as many diverse inputs as possible to the decoder**, in order to prevent overfitting. However, from the perspective of the decoder, this is indeed similar to a many-to-one mapping, as it needs to generate tokens of the same character for these equivalent IDS.
>
> ### Q4: The detailed structure of IDS Encoder
> IDS encoder does not contain a Transformer, in fact its learnable parameters are limited to a simple embedding layer. We apologize for any confusion that may have arisen from the name "encoder".
>
> IDS encoder does not perform complex encoding because **we want to preserve the independence of each element within an IDS**. This also allows the decoder to maintain fine-grained attention on each position of the input sequence.
>
> Specifically, IDS encoder consists of three parts: decomposition, equivalent construction, and embedding.
> Firstly, it recursively breaks down the input IDS into one IDC and corresponding components at each level to figure out whether there is a equivalent. Next, it produces all the equivalents according to certain rules. Finally, it pads to the maximum length and passes through an embedding layer to obtain the final features.
>
> ### Q5: More about the first limitation (fancy and irregular font styles)
> Thank you for your thorough review of our work, and we are happy to discuss and clarify this issue further. We believe that fancy and irregular fonts are difficult to generate due to several reasons:
> 1. As pointed out by the reviewer, one reason is that **novel fonts have a great topological difference from typical ones**, making it challenging for models to learn.
> 2. **The subjective and random nature of font design**. For example, in Figure 9's first font, the "auspicious clouds" decoration appears on every glyph, but its position, proportion, and shape are carefully designed, making it difficult for the model to grasp the regular pattern.
> 3. **Limited reference samples**. More complex styles often require more reference samples to imitate, which requires a trade-off between reference quantity and generation quality.
>
> Although we only show the results of our method in Figure 9, please note that **this limitation is common to all font generation methods**. LF-Font[1] has discussed this issue and CF-Font[2] aims to reduce the difference between content and target styles through content fusion. Since the font in Figure 9 is too novel and lacks generality, we did not include it in the comparison in Figure 5.
>
> We include all methods' results on those novel fonts in Figure R1(in the rebuttal PDF) to support our conclusions. Figure R1 shows that methods based on disentanglement are limited by the content font, and generate worse results when the content style is far away from the target style. Unfortunately, this limitation has not been well addressed yet, we hope to leave it for future works.
>
> [1] Park, S., Chun, S., Cha, J., Lee, B., & Shim, H. (2022). Few-shot Font Generation with Weakly Supervised Localized Representations. In IEEE Transactions on Pattern Analysis and Machine Intelligence: Vol. PP (pp. 1–17). https://doi.org/10.1109/TPAMI.2022.3196675
> [2] Wang, C., Zhou, M., Ge, T., Jiang, Y., Bao, H., & Xu, W. (2023). CF-Font: Content Fusion for Few-shot Font Generation. IEEE/CVF Conference on Computer Vision and Pattern Recognition (CVPR).

---

> > ### Author Response · Authors · 2024-08-10
> > **Result of ablation study requested in Q2**
> >
> > Dear reviewer, we added content glyph as input in addition to the IDS. The experimental results are listed below:
> > |       UFUC&3shot             | FID↓ | L1↓ | LPIPS↓ | RMSE↓ | SSIM↑ |
> > |------------|---------|---|---|---|---|
> > | Both Modalities   | **8.2603**  | **0.1576**  | **0.1316**  | **0.3744**  | **0.4830**  |
> > | Only IDS           | 8.4922  | 0.1597  | 0.1338  | 0.3775  | 0.4782  |
> >
> > As expected, the additional information and parameters bring about an improvement in performance. However, please note:
> > 1. This improvement is not significant compared to the increase in training costs (double the previous amount).
> > 2. The style of the content font will affect the generated results to some extent.
> >
> > Unfortunately, it appears that image uploads are not allowed during the author-reviewer discussion period, we are unable to share the visualization.

---

> > > ### Comment · Reviewer_D5bL · 2024-08-12
> > > **My final recommendation**
> > >
> > > Thank you to the authors for the detailed responses, which have addressed most of my concerns. My final recommendation is a borderline accept.
> > >
> > > The ablation study on dual-modality inputs effectively demonstrates the superiority of IDS, and I now have a clearer understanding of its approach.
> > >
> > > I would suggest incorporating the expanded discussion on limitations into the final version.
> > >
> > > Additionally, including a reference to the paper [1] on vector font synthesis would be beneficial, as it similarly encodes glyphs using sequential descriptions and avoids morphing the content glyph.
> > >
> > > Ref: "Deepvecfont: synthesizing high-quality vector fonts via dual-modality learning." ACM Transactions on Graphics (TOG), 2021.

---

> > > > ### Author Response · Authors · 2024-08-13
> > > > **Thank you for the comments**
> > > >
> > > > Thank you very much for raising the score and for your thoughtful comments. Following your advice, we will add the expanded discussion on limitations to the final version. The mentioned Deepvecfont does indeed take sequences as input and avoid directly morphing the content glyph, we are glad to cite it in our paper.

---

### Official Review · Reviewer_MtRY · 2024-07-13

**Soundness:** 3
**Presentation:** 3
**Contribution:** 3
**Rating:** 6
**Confidence:** 3

**Summary:**

IF-Font introduces a novel approach to few-shot font generation by using Ideographic Description Sequence (IDS) instead of traditional source glyphs to control the semantics of generated glyphs. This method quantizes reference glyphs into tokens and models the token distribution of target glyphs using IDS and reference tokens. IF-Font effectively synthesizes glyphs with neat and accurate strokes, significantly outperforming existing methods in both one-shot and few-shot settings, particularly when the target styles differ from the training font styles. The method redefines font generation as a sequence prediction task, enhancing the quality and consistency of the generated glyphs.

**Strengths:**

1. Novel Paradigm: IF-Font introduces a new approach by using IDS to control glyph semantics, eliminating the need for content-style disentanglement and reducing artifacts.

2. High-Quality Generation: The method excels in producing glyphs with neat and correct strokes, maintaining consistent style even with limited reference glyphs.

3. Cross-Linguistic Capabilities: IF-Font allows for the creation of new and non-existing Chinese characters, demonstrating flexibility and adaptability across different linguistic structures.

**Weaknesses:**

1. The author proposes a content-style disentanglement method where style extraction relies on the decomposition of glyphs. However, this setup may not be necessary. In the fields of diffusion models and generative AI, there are many methods that can extract styles from a single image with minimal content interference, such as IP-Adapter and Instant-Style. The work FontDiffuser, which generates text using diffusion models, does not decompose the text but still achieves excellent style and content disentanglement. Therefore, I have doubts about the advancement and necessity of the method proposed in this paper.

2. The compared methods are relatively old and do not include comparisons with the most advanced methods such as FontDiffuser.
Since the target glyph is reconstructed from a quantized sequence, the accuracy of this reconstruction sets a ceiling on IF-Font's potential performance.

**Questions:**

1. Is there any cherry-picking of results?
2. What is the real usability rate of the proposed method?
3. Besides inference speed, what advantages does the VQ-GAN method have compared to diffusion-based methods?

**Limitations:**

see weaknesses

---

> ### Author Rebuttal · Authors · 2024-08-07
>
> ### Q1: The advancement and necessity of our method
>
> Font generation differs from general image generation tasks. **Our contribution lies in finding a way to describe ideographic characters as "text" and successfully applying it to font generation**, solving the problem of low-quality generation in previous methods.
>
> IP-Adapter and Instant-Style are diffusion-based text-to-image methods. They use text to control the generation and aim to generate results that strictly follow the text prompt while having the style of reference images. However, their target is much more complex than glyphs, and the corresponding information can usually be described in natural language. Their generation conditions are relatively loose, and there are multiple reasonable results, but with the problem of content leakage.
>
> The content of glyphs is just a character, which is difficult to describe with text, so prior works use the glyph image as the input content. Since the glyph used as the input content also has a style, there is a problem of style leakage.
>
> In summary, these two fields have some intersection but are not identical, and the problems they face are different. We believe that our method has outstanding advantages and contributions in font generation.
>
> ### Q2: The comparison with FontDiffuser
> We have included the comparisons with FontDiffuser, which shows that although FontDiffuser performs well, our method is still at an advantage:
>
> | Model(ufsc)            | FID↓ | L1↓ | LPIPS↓ | RMSE↓ | SSIM↑ |
> |------------|---------|---|---|---|---|
> | FontDiffuser (1shot)  |  3.9969 | 0.1938  |  0.1371 |  0.4180 | 0.4076  |
> | Ours (1shot)            | 6.7695  |  0.1529 | 0.1307  | 0.3688  | 0.4915  |
> | FontDiffuser (3shot)  | **3.6979**  |  0.1774 |  0.1248 | 0.3980  | 0.4370  |
> | Ours (3shot)            | 6.8359  |  0.1478 | 0.1258  | 0.3620  | 0.5021  |
> | FontDiffuser (8shot)  |  4.1017 | 0.1748  | 0.1234  | 0.3947  | 0.4420  |
> | Ours (8shot)            | 6.7383  | **0.1429**  | **0.1216**  | **0.3552**  | **0.5140**  |
>
> | Model(ufuc)            | FID↓ | L1↓ | LPIPS↓ | RMSE↓ | SSIM↑ |
> |------------|---------|---|---|---|---|
> | FontDiffuser (1shot)  | 8.2524  | 0.1914  | 0.1527  | 0.4157  | 0.4163  |
> | Ours (1shot)            | 8.4844  | 0.1651  | 0.1387  | 0.3845  | 0.4676  |
> | FontDiffuser (3shot)  | **7.6444**  | 0.1771  | 0.1413  | 0.3981  | 0.4418  |
> | Ours (3shot)            | 8.4922  | 0.1597  | 0.1338  | 0.3775  | 0.4782  |
> | FontDiffuser (8shot)  | 8.9166  | 0.1702  | 0.1367  | 0.3890  | 0.4543  |
> | Ours (8shot)            | 8.3203  | **0.1561**  | **0.1305**  | **0.3728**  | **0.4864**  |
>
> ### Q3: Performance ceiling due to quantization
> Yes, our performance is limited by the reconstruction accuracy of VQ-GAN. However, glyph images are binary and relatively simple, fine-tuning VQ-GAN can minimize the precision loss.
>
> We use original VQ-GAN checkpoints without fine-tuning in order to highlight our main contributions.
>
> ### Q4: Cherry-Picking & Real Usability Rate
> In the paper, we selected representative samples to demonstrate our method's distinguishing features. In fact, almost all generated samples are of high quality, except for those shown in the 4.6 failure cases section.
>
> ### Q5: The advantages of the VQ-GAN method compared to diffusion-based methods
> 1. **Conforms to writing habits**: Using IDS and quantized tokens together in auto-regressive modeling implicitly incorporates writing order.
> 2. **Scalability**: The VQ-GAN method has fewer parameters, and is easier to scale. In addition, it has an advantage in terms of training speed and memory usage, making it suitable for deployment on terminal devices.
> 3. **Robustness**: Due to vector quantization, characters are represented by a limited number of tokens (only 256 types), which reduces the difficulty of modeling for the decoder and makes it less likely to produce artifacts.
>
> As is well known, the diffusion model is excellent, but the focus of this paper is not on a specific model. Indeed, our key idea is model-agnostic, which can be also applied to diffusion or other types of networks.

---

> > ### Comment · Reviewer_MtRY · 2024-08-13
> >
> > Thank you for the detailed response. Due to the debate regarding the paper's motivation and novelty, I am glad to improve my current score to 6. Nevertheless, I hope this work can cite the following paper.
> >
> > 1. CLIPFont: Text Guided Vector WordArt Generation
> > 2. FontDiffuser: One-Shot Font Generation via Denoising Diffusion with Multi-Scale Content Aggregation and Style Contrastive Learning

---

> > > ### Author Response · Authors · 2024-08-13
> > >
> > > Thank you for acknowledging our work. We would be glad to cite the two papers in Section 2 (Related Works) of the final version.
> > >
> > > Just a gentle reminder: We have noticed that the score has not been updated yet (the reviewer mentioned that it would be raised to 6), which may be due to forgetting to save. :D

---

### Author Rebuttal · Authors · 2024-08-07

We attempted our best to address the questions as time allowed. We believe the comments & revisions have made the paper stronger and thank all the reviewers for their help. Please find individual responses to your questions below. The PDF file for the figures is attached to this general response.

---

> ### Comment · Area_Chair_r1Am · 2024-08-08
> **Reviewer-author discussions**
>
> To the authors:
> Thanks a lot for the rebuttal.
>
> To all reviewers:
> Please read the rebuttal and other review comments if necessary, and have a discussion with the authors if you have further comments.
>
> Best,
> AC

---

### Decision · Program_Chairs · 2024-09-25

**Decision:**

Accept (poster)

**Comment:**

Initially, this paper received mixed ratings: 5, 4, 6 and 5. The comments included more experimental comparison, advantages over some methods, clarification and discussions for the proposed method and experiments, and so on.After post-rebuttal discussion, the ratings were changed to 6, 5, 7, and 5. Three reviewers were satified with the rebuttal and upgraded the ratings. The other reviewer was also satisfied with rating and did not update the rating.  The AC read the comments and the paper carefully.